Have female twisted-wing parasites (Insecta: Strepsiptera) evolved tolerance traits as response to traumatic penetration?

Jandausch Kenny 1 2 kenny.jandausch@uni-jena.de
Michels Jan 3
Kovalev Alexander 3
http://orcid.org/0000-0001-9712-7953 Gorb Stanislav N. 3
http://orcid.org/0000-0001-7390-1318 van de Kamp Thomas 4 5
Beutel Rolf Georg 1
Niehuis Oliver 2
http://orcid.org/0000-0002-7090-6612 Pohl Hans 1 hans.pohl@uni-jena.de
1 Institute of Zoology and Evolutionary Research, Friedrich Schiller University Jena , Jena, Thuringia , Germany
2 Department of Evolutionary Biology and Ecology, Albert Ludwig University Freiburg , Freiburg , Germany
3 Functional Morphology and Biomechanics, Zoological Institute, Christian-Albrechts-Universität zu Kiel , Kiel , Germany
4 Institute for Photon Science and Synchrotron Radiation (IPS), Karlsruhe Institute of Technology (KIT) , Eggenstein-Leopoldshafen , Germany
5 Laboratory for Applications of Synchrotron Radiation (LAS), Karlsruhe Institute of Technology (KIT) , Karlsruhe , Germany
Gillespie Joseph
Electronic publication date: 2022 Aug 16
Publication date: 2022
Volume: 10
Electronic Location ID: e13655
Received 2022 Mar 16; Accepted 2022 Jun 9
Copyright: © 2022 Jandausch et al.
Copyright year: 2022
Copyright holder: Jandausch et al.
License: This is an open access article distributed under the terms of the Creative Commons Attribution License, which permits unrestricted use, distribution, reproduction and adaptation in any medium and for any purpose provided that it is properly attributed. For attribution, the original author(s), title, publication source (PeerJ) and either DOI or URL of the article must be cited.
License URL: https://creativecommons.org/licenses/by/4.0/

Keywords: Micro-indentation, Confocal laser scanning microscopy, Resilin, Female tolerance traits, Interspecific competition

Funding: German Research Foundation (DFG) NI 1387/9-1; PO 1207/4-1; GO 995/46-1; BE 1789/15-1 This work was founded by the German Research Foundation (DFG) (NI 1387/9-1; PO 1207/4-1; GO 995/46-1; BE 1789/15-1). The funders had no role in study design, data collection and analysis, decision to publish, or preparation of the manuscript.

==============================
Traumatic insemination describes an unusual form of mating during which a male penetrates the body wall of its female partner to inject sperm. Females unable to prevent traumatic insemination have been predicted to develop either traits of tolerance or of resistance, both reducing the fitness costs associated with the male-inflicted injury. The evolution of tolerance traits has previously been suggested for the bed bug. Here we present data suggesting that tolerance traits also evolved in females of the twisted-wing parasite species Stylops ovinae and Xenos vesparum. Using micro-indentation experiments and confocal laser scanning microscopy, we found that females of both investigated species possess a uniform resilin-rich integument that is notably thicker at penetration sites than at control sites. As the thickened cuticle does not seem to hamper penetration by males, we hypothesise that thickening of the cuticle resulted in reduced penetration damage and loss of haemolymph and in improved wound sealing. To evaluate the evolutionary relevance of the Stylops-specific paragenital organ and penis shape variation in the context of inter- and intraspecific competition, we conducted attraction and interspecific mating experiments, as well as a geometric-morphometric analysis of S. ovinae and X. vesparum penises. We found that S. ovinae females indeed attract sympatrically distributed congeneric males. However, only conspecific males were able to mate. In contrast, we did not observe any heterospecific male attraction by Xenos females. We therefore hypothesise that the paragenital organ in the genus Stylops represents a prezygotic mating barrier that prevents heterospecific matings.

Introduction

Sexual reproduction and copulation occur in many different varieties across the animal kingdom. One of the most bizarre forms of sexual interaction is traumatic mating, which involves the injury of one sexual partner during copulation. In one particular form of traumatic mating, traumatic penetration (Lange et al., 2013), the female gets injured during the mating process, but receives no sperm. In contrast, traumatic mating resulting in the injection of sperm and the insemination of the female is referred to as traumatic insemination (Lange et al., 2013). Traumatic insemination occurs in different groups of animals, such as flatworms, snails and slugs, annelid worms, and arthropods (Lange et al., 2013; Tatarnic, Cassis & Siva-Jothy, 2014; Brand, Harmon & Schaerer, 2022). However, one of the best known and studied organisms in this context is the bed bug (Cimex lectularius) (Michels, Gorb & Reinhardt, 2015; Reinhardt, Anthes & Lange, 2015; Brand, Harmon & Schaerer, 2022), in which traumatic insemination occurs within a paragenital organ: the spermalege.

Traumatic insemination leads to injuries of females and is recognised as an example of sexual conflict (Lange et al., 2013; Tatarnic, Cassis & Siva-Jothy, 2014). Females unable to avoid unnecessary mating are predicted to likely evolve resistance traits or tolerance traits that reduce the fitness costs associated with sexual interaction with males. Resistance traits, which reduce the fitness of the copulating male(s), can result in a co-evolutionary arms race. Such arms races are generally thought to lead to accelerated trait exaggeration, such as the formation of species-specific differences in copulatory organs (Arnqvist & Rowe, 2002; Parker, 2006; Lange et al., 2013). Michels, Gorb & Reinhardt (2015) found evidence for female tolerance traits (i.e., traits that reduce the fitness costs associated with a sexually conflicting interaction of the female without decreasing the fitness of the male) in the form of resilin in the spermalege in bed bugs. The elastomeric protein resilin seals the sexually imposed wounds and physically facilitates copulation by males (Michels, Gorb & Reinhardt, 2015, p. 6).

The endoparasitic insect order Strepsiptera comprises ca. 600 described species. It is characterised by numerous derived characters of all life stages and in both sexes (Pohl & Beutel, 2008). All species of the order display extreme sexual dimorphism. The males are free-living (Figs. 1A, 1C); the only function of their extremely short life span of a few hours is to find females and to mate. The females are usually obligatory endoparasites of other insects, in which they stay during most of their larval development and as adults (Figs. 1B, 1D). Only females of the family Mengenillidae are a notable exception, as they are free-living in the adult stage. Female Strepsiptera are wingless and structurally strongly simplified, as compared to the males. Their genital apparatus is extremely reduced: ovaries, vagina, receptacula seminis, genital chamber, bursa copulatrix, and accessory glands are missing, and the eggs float freely in the hemolymph. A single birth organ is present in females of Mengenillidae, with an opening in the region of sternite VII through which the minute (ca. 200–250 µm) primary larvae are released.

Figure 1 Photographs of Stylops ovinae and Xenos vesparum.

(A) Frontal view of an adult male S. ovinae. (B) One female of S. ovinae protrudes from its host’s metasoma (Andrena vaga). (C) Frontal view of an adult male X. vesparum. (D) Two females of X. vesparum protruding from their host’s metasoma (Polistes dominula). Arrowheads indicate strepsipteran females.

The genera Stylops and Xenos investigated in the present study belong to the clade Stylopidia, whose members utilise only pterygote insects as hosts and whose females are obligatory endoparasites. Females of Stylopidia are characterised by secondary tagmosis: head, thorax, and the anterior part of abdominal segment I form a compact cephalothorax, while antennae, compound eyes, and legs are missing. The large sack-shaped posterior portion of the body remains within the host’s abdomen, whereas the cephalothorax is exposed. A secondary copulation opening is located in the exposed cephalothorax. On the ventral side of the cephalothorax, between head and prosternum, a birth opening is present in the majority of species. This is the external opening of the brood canal, which is connected with the birth organs in the abdomen. The birth opening enables the primary larvae to leave the females and is used in most species for insemination (Stylopiformia). In Corioxenidae, insemination occurs either in the region of the mouth opening or in the membranised pleural region (Pohl & Beutel, 2008). The cuticle of the female is three-layered, as the female does not shed its larval exuviae. The cuticle layers are detached in the ventral area and form the brood canal. The exuvia of the second larval stage is strongly sclerotised in the cephalothoracic region, thinning only at the birth opening. The exuvia of the tertiary larva is extremely thin and weakly sclerotised (Figs. 2A, 2B).

Figure 2 Sagittal sections of female cephalothoraces of Stylops ovinae (A) and Xenos vesparum (B).

Abbreviations: bc, brood canal; bo, birth opening; ct, cuticle; ed, epidermis; ex2, exuvia of the secondary larval stage; ng, Nassonow’s gland; po, paragenital organ; ws, wounding site; cs, control site. (A) Modified from Peinert et al. (2016). (B) Modified from Richter et al. (2017).

The endoparasitic lifestyle of female Stylopidia has strongly influenced the mating strategy and the mating behaviour of its species, as only the female’s cephalothorax protrudes from the host’s abdomen (Figs. 1B, 1D). In S. ovinae, traumatic insemination takes place at a paragenital organ (po) located in front of the birth opening (Peinert et al., 2016) (Figs. 2A, 3A), while in X. vesparum, traumatic insemination occurs at the anterior part of the brood canal (bc) (Figs. 2B, 3B). However, Beani et al. (2005) discussed an alternative sperm route in X. vesparum, namely by release of sperm into the brood canal: the sperm could then reach the hemocoel of the female via the birth organs.

Figure 3 Female cephalothoraces of Stylops ovinae (A) and Xenos vesparum (B) with mating signs (outer cuticle of the cephalothorax removed).

Abbreviation: ms, mating sign. (A) Modified from Peinert et al. (2016).

Previous studies have shown that the cuticle of the paragenital organ of S. ovinae and the cuticle of the anterior brood canal of X. vesparum is about three times thicker than the cuticle in spatial proximity (Löwe, Beutel & Pohl, 2016; Peinert et al., 2016; Richter et al., 2017). However, the material composition of the cuticle at specific penetration sites has not been studied and compared to that of the surrounding areas. This information could give additional clues whether Strepsiptera have evolved resistance traits or tolerance traits in response to traumatic penetration.

The penises of different species of Stylops vary greatly in shape and size, whereas those of males of Xenos differ almost exclusively in size (Fig. 4). Variation in penis shape, as reported for Stylops, has so far only been described from the genus Caenocholax (Myrmecolacidae) (Kathirithamby et al., 2015). The varying shape and size of the intromittent organs of different Stylops species could be either the result of intersexual co-evolution or avoidance of interbreeding between sympatric occurring congeners. Following Eberhard (2004a), females encountering males may be passively protected via species-specific pheromones, for example by decreasing the chance of interspecific mating.

Figure 4 Penises of different species of Stylops (red) and Xenos (yellow).

(A) S. ovinae. (B) S. hammella. (C) S. liliputanus. (D) S. aterrimus. (E) S. melittae. (F) X. columbiensis. (G) X. moutoni. (H) X. oxydontes. (I) X. vesparum. (B–E) Modified from Kinzelbach (1978), (F) modified from Cook, Mayorga-Ch & Sarmiento (2020), (G) modified from Kifune & Maeta (1985), (H) modified from Nakase & Kato (2013).

It is interesting to note that several of the 67 described Stylops species (Straka, Jůzová & Nakase, 2015) occur in sympatry and can be encountered in the same habitat at the same time of the year (e.g., at least three different Stylops species have been recorded flying in March in the vicinity of Jena; H. Pohl, 2020, personal observations). In contrast, out of the 32 described Xenos species, only very few occur in sympatry (e.g., only two species in the Western Palearctic, one likely restricted to Libya, while the other one is widespread). Furthermore, Eberhard (2004b) concluded that species-specific traits in males are missing when females lack specific contact organs associated with copulation. We hypothesise that the paragenital organ of Stylops is such a specific contact organ, which suggests that the male genital armature will be more species-specific in this genus. To date it remains unclear whether heterospecific copulations or heterospecific attraction of Strepsiptera occurs in nature and if so, whether the species-specific morphology of the females’ paragenital organs pose a significant prezygotic reproductive barrier that prevents penetration by heterospecific males.

We combined observational and experimental methods to address the question of whether Stylops and Xenos females evolved resistance traits or tolerance traits as counter-adaptations against traumatic wounding. We used confocal laser scanning microscopy to evaluate the material composition of the cuticle at the wounding sites of S. ovinae (paragenital organ) and X. vesparum (anterior brood canal) as well as at surrounding areas. We specifically assessed the presence of resilin, a soft and elastic protein found in large proportions in the cuticle of the spermalege of female bed bugs (Michels, Gorb & Reinhardt, 2015). We then measured the force required to penetrate the cuticle of the paragenital organ of S. ovinae and the anterior brood canal of X. vesparum using micro-indentation. We compared the results to the force necessary to penetrate the cuticle in control areas.

To clarify the question why the penises of different Stylops species strongly differ in size and shape, but not those of Xenos, we used field experiments to assess whether virgin females of both species attract heterospecific males. We subsequently used laboratory mating experiments to test whether heterospecific Stylops males are able to anchor and penetrate the female paragenital of S. ovinae with their penis. Additionally, we reconstructed the three-dimensional morphology of the penises and the female paragenital organ in Stylops using X-ray computed tomography to assess the morphological fit of the two organs. Positive correlation of shape and size of penises and paragenital organs of different Stylops species may indicate an interspecific copulatory barrier comparable to a lock-and-key mechanism. Finally, we used 3D models of penises of S. ovinae and X. vesparum to estimate the extent of intraspecific variation using geometric morphometrics. Comparing the magnitude of intraspecific penis variation of both investigated species could indicate whether this contributed to the higher diversification of penis morphology on the genus level.

Material and Methods

Studied insects

A total of 70 Andrena vaga (Hymenoptera, Andrenidae) parasitised by S. ovinae were collected in Langerwehe (North Rhine-Westphalia, Germany) (February 16, 2020, and March 11, 2021, by E. Holtappels, K. Jandausch, H. Pohl, and D. Tröger). During transport and preparation of the experiments, the bees were kept dark in glass vessels (0.5 L) half filled with moist sand at ~4 °C to prevent males of S. ovinae from hatching.

The hosts of X. vesparum, Polistes dominula (Hymenoptera, Vespidae), were collected in Mettenheim (Rhineland-Palatinate, Germany) (July 1, 2018; July 21, 2020, and August 11, 2021, by K. Jandausch, H. Pohl and D. Tröger) on a trumpet creeper bush (Campsis radicans). Parasitised P. dominula are attracted to trumped creeper bushes and feed on extrafloral gland secretions (Beani et al., 2018). Wasps collected in 2018 and 2021 were used for the attraction experiments (see below). A total of 137 wasps were collected in 2020 for the micro-indentation experiments. In the laboratory, each wasp was assigned to one of four groups: wasps with extruded female X. vesparum (n = 32), wasps with extruded male puparia (n = 28), wasps with extruded females and male puparia (n = 3), and wasps without externally visible infestation (n = 40). Only one male puparium was empty, indicating that the majority of females were unfertilised. All parasitised wasps were kept in small groups (n = 5–8) in glass vessels (0.5 L) covered with gauze at room temperature and were fed ad libitum with water and diluted honey. Wasps without externally visible infestation were kept in an “aerarium” (40 cm × 40 cm × 60 cm) (Papa Papillon, Bern, Switzerland) and checked every day for freshly extruded X. vesparum.

One female specimen of S. melittae (ID: SFl87) and one of S. hammella (ID: SFc12) preserved in ethanol were provided by Jakub Straka.

Confocal laser scanning microscopy

Cephalothoraces of a virgin female S. ovinae and of a virgin female X. vesparum were dissected, transferred to glycerol (≥99.5%, free of water, two times distilled; Carl Roth GmbH & Co. KG, Karlsruhe, Germany) and mounted in glycerol on object slides with high-performance cover slips (Carl Zeiss Microscopy GmbH, Jena, Germany) as described earlier (Michels & Büntzow, 2010; Michels & Gorb, 2012). Four autofluorescences exhibited by the cephalothoracic cuticle structures were visualised with a ZEISS LSM 700 confocal laser scanning microscope (Carl Zeiss Microscopy GmbH, Jena, Germany) using four solid-state lasers and bandpass and longpass emission filters as previously described (Michels & Gorb, 2012; Michels, Gorb & Reinhardt, 2015). The microscope system was controlled by the software ZEISS Efficient Navigation 2009 (ZEN 2009; Carl Zeiss Microscopy GmbH, Jena, Germany). A ZEISS Plan-Apochromat lens with a numerical aperture of 0.45 (Carl Zeiss Microscopy GmbH) was applied. Using ZEN 2009, a maximum intensity projection (MIP) was created from each of the obtained data sets. The four different autofluorescence visualisation results shown on the MIPs were colour-coded and overlaid as described earlier (Michels & Gorb, 2012). The resulting micrographs show differences in the autofluorescence composition of the analysed cuticle structures and indicate differences in the material composition of these structures. Red cuticle structures consist mainly of strongly sclerotised chitinous material, green ones are chitinous and are weakly sclerotised and membranous and either weakly sclerotised or non-sclerotised, and blue ones contain large proportions of resilin (Michels & Gorb, 2012).

Specimen preparation

Living females were extracted from their hosts with fine tweezers immediately before the micro-indentation experiments. The strongly sclerotised outer cuticle of the cephalothorax was removed to gain access to the penetration sites. The cephalothoraces were then fixed with micro-needles on a silicon block (Silicone HR-N; Reckli, Herne, Germany). Two types of fixations were used when handling S. ovinae: to make the ventral wall of the paragenital organ (wounding site) accessible, the females were fixed with the morphological dorsal side directed upwards, and the head area of the female was folded back. To expose the anterior part of the brood canal (control site), females were fixed with the ventral side directed upwards. The brood canal was chosen as the control site for S. ovinae, as it resembles the location where traumatic penetration takes place in other families of Strepsiptera (e.g., Xenidae). Females of X. vesparum were fixed exclusively with the ventral side directed upwards, as wounding and control sites are located on the same side. At the wounding sites, micro-indentation in S. ovinae and in X. vesparum was carried out in the areas where mating signs are located in mated females (Fig. 3). In X. vesparum, reference measurements at control sites were carried out at the posterior end of the brood canal where the cuticle is much thinner, representing the general state of the cuticle at the cephalothorax. All preparations were performed using a stereomicroscope (Olympus SZX12; Olympus, Tokyo, Japan).

Micro-indentation experiments

Micro-indentation experiments were carried out on living virgin females without mating signs. We studied 24 females of S. ovinae (wounding site: 14; control site: 10) and 12 females of X. vesparum. In X. vesparum, we performed both measurements on the same females. The force to penetrate the cuticle of the females was measured by inserting steel micro-needles with diameters of 5.6 µm when studying S. ovinae and 4.1 µm when studying X. vesparum. In comparison, the tips of the penises of S. ovinae measured on average 2.8 µm (n = 3, min. 2.6 µm, max. 3.0 µm), and those of X. vesparum measured on average 0.8 µm (n = 3, min. 0.7 µm, max. 0.9 µm). All tips were analysed using scanning electron micrographs of the penises and tips of the micro-needles. Each micro-needle was directly glued to a 10-g force transducer (World Precision Instruments, Sarasota, FL, USA) with cyano-acrylate glue (Ergo 5925 Elastomer; Kisling AG, Wetzikon, Switzerland). The force transducer was attached to a motorised micro-manipulator. To perform the indentation experiments, the force transducer was moved down with 200 µm/s velocity. During the experiments, the females were moistened with a drop of tap water to prevent drying artefacts. Each specimen was penetrated several times on slightly different positions. The whole system was connected to a computer running the software AcqKnowledge 3.7.3 (Biopac System Inc., Goleta, CA, USA) (Fig. S2). This software was used to record and process the measured force and time/travelled distance. All measurements were controlled visually with a stereomicroscope (LEICA MZ 12.5, Wetzlar, Germany) to guarantee the penetration on the chosen location. Individual measurements were documented using a video camera (Basler piA1900-32g; Basler Vision Technologies, Ahrensburg, Germany) attached to the stereomicroscope. The software SteamPix5 (Norpix Inc., Montreal, QC, Canada) was used to record the videos.

Scanning electron microscopy

Scanning electron micrographs of the penises of S. ovinae and of X. vesparum were taken with a Philips ESEM XL30 (Philips, Amsterdam, Netherlands) (Fig. S3). The same equipment was used to obtain micrographs of the tips of the steel microneedles used in the micro-indentation experiments. The penises were air-dried, glued to a microneedle, fixed on a rotatable specimen holder (Pohl, 2010), and sputter-coated with gold using an Emitech K500 (sample preparation division, Quorum Technologies Ltd., Ashford, England).

Attraction experiments

In 2020, four bees, and in 2021, eight bees from Langerwehe parasitised with S. ovinae females were kept in an “aerarium” (see above). The “aerarium” was placed in the field and orientated in the direction of the wind flow to increase the dispersion of female pheromones. The attraction experiments were conducted on March 18 and 19, 2020, and between March 24 and March 26 and on March 29, 2021, in the vicinity of Jena (Thuringia, Germany). Approaching males were collected from the gauze of the “aerarium” with an aspirator. In 2020, two, and in 2021, 40 of the attracted males were taken to the lab alive in snap-cap vials. To keep the males vital for the mating experiments, they were transported in a cold bag at ~5 °C. The remaining males were immediately fixed in 100% ethanol.

An identical setup was used to attract male X. vesparum in the backyard of the Phyletisches Museum Jena (Thuringia, Germany). On July 18, 2018, five, and on August 16, 2021, eight infected wasps from Mettenheim were placed in an “aerarium”. Beani et al. (2018) carried out similar attraction experiments with parasitised wasps in small vials to attract males of X. vesparum. In contrast to S. ovinae, males of X. vesparum were fixed in 70% ethanol after collecting them from the gauze, as no further mating experiments were carried out with this species.

Species identification of the attracted males

To identify Stylops species, we analysed a 605-bp-long fragment of the mitochondrial gene COI in all males collected in 2020 and in the four males that we used in the mating experiment in 2021. The DNA was extracted from legs using the QIAGEN QIAamp DNA Micro, following the protocol of the manufacturer. PCR amplification was carried out with the oligonucleotide primers 5′-TCW ACA AAT CAT AAA ATA ATT GG-3′ (CO122For), 5′-TCC TCC TCC TAA AGG RTC RAA-3′ (CO16669Rev), 5′-TWT CWA CHA AYC ATA ARG ATA TTG G-3′ (Cox1LCO_DEG) and 5′-TCA ATT TCC AAA YCC YCC YAT-3′ (Cox1ALEX_DEG) published by Folmer et al. (1994), McMahon, Hayward & Kathirithamby (2009), and Jůzová, Nakase & Straka (2015). PCRs were performed with the Invitrogen Taq DNA Polymerase for standard PCR (Thermo Fisher Scientific Inc., Waltham, MA, USA) including dNTPs, PCR buffer, MgCl2, and Taq Polymerase. Applied primers were manufactured by Metabion GmbH (Munich, Germany). PCRs started with a 180 s initial phase at 94 °C, followed by 30 cycles of 45 s at 94 °C, 30 s at 50 °C and 90 s at 72 °C, and ended with one final extension at 72 °C for 10 min. Products were purified using ExoProStar 1-Step (Global Life Sciences Solutions USA LLC, Marlborough, MA, USA). For direct bidirectional Sanger sequencing, samples were sent to Macrogen (Amsterdam, Netherlands). After removing the primer binding sites, forward and reverse sequences of each specimen were aligned using Geneious prime 2021.0.3 and compared to reference sequences for the genus Stylops established by Jůzová, Nakase & Straka (2015) (Fig. S4). All other males were determined by comparing their penis to the penises of the barcoded males.

The attracted Xenos males were identified with the key provided by Kinzelbach (1978).

Mating experiments

In order to determine whether the attracted males are able to insert their penis into the paragenital organ and anchor themselves within virgin females of S. ovinae, we followed the method established by Peinert et al. (2016). The mating experiments were carried out directly after collecting the specimens in the field and returning with them to the lab to ensure the vitality of the short-lived males. The copulations were initiated in transparent plastic trays (4 cm diameter, 1 cm high) at 21 ± 1 °C. The metasoma of each parasitised host was removed and was attached directly to modelling clay with its anterior end on the bottom of the trays. The males were placed one at a time in plastic dishes, and the dishes’ opening were closed with a transparent lid to prevent the males from escaping. After about 2 min, when each male had mounted the metasoma of the host bee and attempted to mate with the female, the males were removed and fixed in 100% ethanol for later species identification unless otherwise stated. Note that we determined all males after the experiments, either by barcoding or by comparing the penises of the males with those of barcoded males. Mating attempts were recorded with a Canon EOS 7D digital SLR equipped with a Canon MP-E 65 mm macro lens (Canon, Krefeld, Germany) or an Apple iPhone SE (second generation) (Cupertino, CA, USA) through the eyepiece of a Leica MS 5 stereomicroscope (Leica, Wetzlar, Germany). We used a cold light source (KL 750; Schott, Mainz, Germany) as lighting.

In total, we introduced 18 of the attracted males to females of S. ovinae and tested whether they tried to mate or not. The following mating experiments were performed with virgin females of S. ovinae as described in the preceding paragraph: one male of S. hammella with two females (video recording), and one male of S. ovinae with two females. This latter male mated with one of the females and was fixed in copula with 100% ethanol cooled to −80 °C for a µCT scan. Eleven males of S. melittae were successively placed to one female (video recording). To detect possible injury to females by these heterospecific males during their mating attempts, four males of S. melittae were each placed to four infested host metasomas. Of these, two host metasomas were parasitised by one, one by two, and one by three virgin females. Since the penetration site is marked by a melanised spot on the cuticle 1 day after mating, we stored these females in their host metasoma for 48 h after the mating attempts in the refrigerator at 5 °C. To document the mating signs, we extracted the females from the host, removed the outer cuticle of the cephalothorax, dehydrated the specimens in an ascending ethanol series, and mounted the cephalothorax in Euparal (Carl Roth, Karlsruhe, Germany) on microscope slides. Photographs were taken with a Canon EOS 7D digital SLR equipped with a Mitutoyo M Plan Apo 10x lens (Mitutoyo, Kawasaki, Japan). The slides were illuminated with two flashlights (Yongnuo Photographic Equipment, Shenzhen, China).

X-ray computed tomography and 3D-reconstruction

One female and one male of S. hammella, one male of S. melittae, and one pair of S. ovinae fixed in copula (see mating experiments) were scanned in pure ethanol at the Imaging Cluster at the KIT Synchrotron Radiation Facility using a polychromatic X-ray beam produced by a 1.5 T bending magnet spectrally filtered by 0.5 mm Al. A fast indirect detector system was employed, consisting of a 13 µm LSO:Tb scintillator (Cecilia et al., 2011), a diffraction limited optical microscope (Optique Peter, Lentilly, France) (Douissard et al., 2012), and a 12-bit pco.dimax high speed camera with 2,016 × 2,016 pixels. Scans were done by taking 3,000 projections at 70 fps over an angular range of 180°. An optical magnification of 10× resulted in an effective pixel size of 1.22 µm. The control system concert (Vogelgesang et al., 2016) was employed for automated data acquisition and online reconstruction of tomographic slices for data quality assurance. Data processing included flat field correction and phase retrieval of the projections based on the transport of intensity equation (Paganin et al., 2002). X-ray beam parameters for algorithms in the data processing pipeline were computed by syris (Faragó et al., 2017). The execution of the pipelines, including tomographic reconstruction, was performed by the UFO framework (Vogelgesang et al., 2012). One female of S. melittae was scanned in a SkyScan221 micro-CT (FSU Jena) with beam strength of 40 kV and 300 μA. In a 360° scan, pictures were taken every 0.2° with an exposure time of 5,800 ms. A pixel size of 1.22 μm was achieved. We segmented tomographic data using Dragonfly 4.1 for Windows (Object Research Systems (ORS) Inc, Montreal, QC, Canada, 2019) and used VGStudiomax 2.0.5 (Volume Graphics, Heidelberg, Germany) for visualization and rendering.

Geometric morphometrics

To estimate the intraspecific variation, penises of 18 specimens of S. ovinae (Niedringhaussee, Lower Saxony, Germany) and of 17 specimens of X. vesparum (Jena, Thuringia, Germany) were carefully removed with fine tweezers and dried at the critical point with a Emitech K850 Critical Point Dryer (Sample preparation division, Quorum Technologies Ltd., Ashford, England). Penises were transferred into a plastic pipette tip and scanned in a SkyScan221 micro-CT (FSU Jena) with beam strength of 40 kV and 200 μA. In a 360° scan, pictures were taken every 0.2° with an exposure time of 2,300 ms. A pixel size of 0.7 μm was achieved. Segmentation was performed with Dragonfly 4.1 for Windows (Object Research Systems Inc, Montreal, QC, Canada, 2019). Exported stl files were smoothed and the polygons were reduced and rendered with Blender before exportation as obj files (Blender Foundation, Amsterdam, Netherlands). Landmarks and semilandmarks were placed with Stratovan Checkpoint (Stratovan Cooperation, Davis, CA, USA).

Landmarks

Nine landmarks and 126 semilandmarks on two curves were placed on the 3D objects to describe the overall size and outline shape of the penises. Landmark 1 describes the proximal edge of the phallotreme. Landmark 2 was placed at the deepest point of the ventral angle of the acumen (Fig. S3). Landmark 3 represents the transition from the flat distal part of the penis to the broader base, and landmark 4 marks the posterior edge of the sclerotised penis base. Along these first four landmarks, 54 semilandmarks where projected to represent the posterior outline of the penis shape in detail. Landmark 5 was placed at the distal edge of the phallotreme, followed by landmark 6 at the tip of the acumen. Landmark 7 marks the maximum point of the dorsal curvature of the entire acumen. Landmark 8 represents the dorsal spine. Landmark 9 represents the anterior edge of the sclerotised penis base. A curve consisting of 72 semilandmarks was placed between landmark 5 to landmark 9 to describe the shape of the acumen and the anterior outline.

Statistical analysis

We conducted a total of 146 penetration force experiments (S. ovinae wounding site: 43; S. ovinae control site: 41; X. vesparum wounding site: 38; X. vesparum control site: 36). We analysed 30 measurements from the wounding site of S. ovinae, 40 measurements from the control site of S. ovinae, 38 measurements from the wounding site of X. vesparum, and 36 measurements from the control site of X. vesparum.

For statistical analysis and significance tests, we used the software RStudio (R Core Team, Auckland, New Zealand). Boxplots and raw diagrams were drawn with the same software. We applied the Shapiro-Wilk test for testing for normal distribution and F test for testing for equality of variances among independent datasets. The level of significance was checked with a Kruskal-Wallis test and with a Wilcoxon pairwise comparison (including Bonferroni-Holm correction). In case of the data of X. vesparum, we used Levene’s test for assessing the equality of variances instead of applying an F test, because the data did not follow a normal distribution (Shapiro-Wilk test: p = 0.03403307 [wounding site] and p = 0.01891762 [control site]). The data of S. ovinae, however, were normally distributed (Shapiro-Wilk test: p = 0.09288335 [wounding site] vs. p = 0.7888944 [control site]).

Geometric morphometric analysis was carried out with RStudio (R Core Team, Auckland, New Zealand) and the geomorph package (Adams et al., 2021; Baken et al., 2021). General Procrustes analysis was performed using the gpagen function with semilandmarks allowed to slide between landmarks by minimizing bending energy (define.sliders). Resulting centroid size for each specimen was used for calculation of the coefficient of variance. PCA of the Procrustes shape variables was performed with the function gm.prcomp and the results were visualized with the packages pca3D and rgl to explore major aspects of the geometric variation.

Image processing

All images were processed with Adobe Photoshop Version 21.2.1 (Adobe Systems Incorporated, San Jose, CA, USA). We used Adobe Illustrator Version 24.2.1 (Adobe Systems Incorporated, San Jose, CA, USA) for labelling the plates and for drawings and processing diagrams.

Results

Confocal laser scanning microscopy

Major parts of the cephalothoracic cuticle of both investigated species contain large proportions of resilin, including the areas where traumatic insemination takes place (Fig. 5). The composition of the cuticle in the control areas does not differ from sites where traumatic insemination takes place. In S. ovinae, the spiracles and the main tracheal stems of the cephalothorax consist of sclerotised chitinous material (Fig. 5A). In contrast, different areas on the ventral side of the cephalothorax are not or only weakly sclerotised. These include individual structures of the cephalic area, patches of the lateral pro-, meso-, and metathoracic regions, as well as microtrichia on the surface of abdominal segment I (Fig. 5A). The microtrichia covering almost the entire ventral cephalothoracic surface of X. vesparum are apparently slightly more sclerotised than those on the cephalothorax of S. ovinae (Fig. 5B). However, the spiracles, the main tracheal stems, and the lateral regions of the cephalothorax of X. vesparum are less sclerotised than corresponding areas in S. ovinae.

Figure 5 Confocal laser scanning micrographs (maximum intensity projections) showing the cuticle autofluorescence composition of female Stylops ovinae and Xenos vesparum cephalothoraces (outer cuticle of the cephalothorax removed).

(A) Morphological ventral side of S. ovinae. (B) Morphological ventral side of X. vesparum. Blue structures contain large proportions of the elastomeric protein resilin. Green structures are chitinous and membranous and either weakly-sclerotised or non-sclerotised. Abbreviations: cs, control site; po, paragenital organ; ws, wounding site.

Micro-indentation experiments

The force required to penetrate the paragenital organ of S. ovinae (wounding site; see Fig. 5A) with a micro-needle was statistically significantly higher (mean: 9.732 mN) than the force required at a control site at the brood canal (see Fig. 5A) (mean: 4.578 mN) (Wilcoxon pairwise comparison; p = 2.1e−08*; Tables 1 and 2). Likewise, the force required to penetrate the anterior brood canal (wounding site; see Fig. 5B) of X. vesparum was significantly higher (mean: 3.669 mN) than the force required at the posterior control site in the brood canal (see Fig. 5B) (mean: 2.003 mN) (Wilcoxon pairwise comparison; p = 1.5e−08*; Tables 1 and 2). The absolute forces required to penetrate the cuticle were consistently higher in S. ovinae than in X. vesparum.

Table 1 Means, standard deviations (SD) and p values from Shapiro-Wilk tests (SWT).

	Penetration force	Critical stress	
	Mean (mN)	SD	SWT	Mean (GPa)	SD	SWT	
S. ovinae (wounding site)
n = 30	9.723	3.498	0.09288335	0.402	0.138	0.09427896	
S. ovinae (control site)
n = 40	4.578	1.483	0.7888944	0.186	0.060	0.7888944	
X. vesparum (wounding site)
n = 38	3.669	1.011	0.03403307	0.273	0.075	0.03404492	
X. vesparum (control site)
n = 36	2.003	1.077	0.01891762	0.149	0.080	0.01892339	
Note:

mN, millinewton; GPa, gigapascal.

Table 2 Results of the Wilcoxon pairwise comparison of means for the penetration force and the critical stress of female Stylops ovinae and Xenos vesparum.

Penetration force	S. ovinae (wounding site)	S. ovinae (control site)	X. vesparum (wounding site)	
S. ovinae (control site)	p = 2.1e−08*	–	–	
X. vesparum (wounding site)	p = 2.8e−11*	p = 0.00159*	–	
X. vesparum (control site)	p = 1.6e−13*	p = 5.5e−11*	p = 1.5e−08*	
Critical stress			–	
S. ovinae (control site)	p = 7.5e−09*	–	–	
X. vesparum (wounding site)	p = 4.1e−05*	p = 1.2e−06*	–	
X. vesparum (control site)	p = 1.7e−09*	p = 0.0327*	p = 2.7e−08*	
Notes:

* Indicates significance.

p, probability value.

Critical stress (force per unit area) was calculated to compare the mechanical impact of the intromittent devices on female structures in both species, independent of differences in the diameter of the device. The highest mean value (0.402 GPa) was measured at the wounding sites of S. ovinae (Fig. 6, Table 1). The mean critical stress at the corresponding control sites (0.186 GPa) proved to be statistically significantly lower (Wilcoxon pairwise comparison; p = 7.5e−09*) (Tables 1 and 2). The critical stress value was significantly higher than critical stress values obtained for measurements at wounding and at control sites of X. vesparum (Fig. 6, Tables 1 and 2). The critical stress at the wounding site of X. vesparum (mean value of 0.273 GPa) was significantly higher than that at the control site (mean value of 0.149 GPa) (Wilcoxon pairwise comparison; p = 2.7e−08*) (Tables 1 and 2).

Figure 6 Critical stress values for wounding and control sites of Stylops ovinae (red) and Xenos vesparum (yellow).

Stylops ovinae: wounding site–ventral wall of the paragenital organ, control site–anterior brood canal. Xenos vesparum: wounding site–anterior brood canal; control site–posterior brood canal. Boxes represent the interquartile range between first and third quartiles, and the line inside represents the median. Whiskers denote the lowest and highest values within 1.5× interquartile range from the first and third quartiles, respectively. Circles represent outliers beyond the whiskers.

Attraction experiments

During two consecutive days in 2020, we were able to attract a total of 18 Stylops males in the field by exposing four Andrena vaga parasitised with females of S. ovinae. Using DNA barcoding and/or morphology (i.e., the species-diagnostic shape of the penis), we identified the 18 captured Stylops males as S. hammella (two individuals), S. melittae (13 individuals) and S. ovinae (three individuals). During four consecutive days in 2021, we attracted an additional 94 Stylops males by exposing eight A. vaga parasitised with females of S. ovinae. Of these, 91 were identified as S. melittae and one as S. ovinae. Two specimens were accidentally crushed during capture and remained unidentified. In 2018, we were able to collect 104 males of X. vesparum during a single day, attracted by a single P. dominula female parasitised by a single X. vesparum female. In 2021, we additionally collected 81 males of X. vesparum by exposing 10 P. dominula, each parasitised by X. vesparum females during a single day. Note that each of the above females of S. ovinae and X. vesparum attracted multiple males simultaneously.

Mating experiments

All Stylops males of the attraction experiments, irrespective of their species identity, mounted the parasitised host bee and searched for a suitable position on the bee’s metasoma to mate with the S. ovinae females. The male of S. hammella was unable to insert its penis into the paragenital of two female S. ovinae and to anchor itself (Video S1). All S. melittae males were able to insert their penises into the paragenital organ, but they were unable to anchor (Video S1). In three out of six copulation attempts of S. melittae males with S. ovinae females, the female’s cuticle was punctured. However, none of the resulting scars in the cuticle were found at the terminal end of the paragenital organs, the only location penetrated by conspecific males (H. Pohl and K. Jandausch, 2020, personal observations), but rather in the anterior region of the paragenital organ (Fig. S1). Only the conspecific male was able to perforate the terminal part of the female’s paragenital organ (fixed in copula, µCT scan).

Co-adaptation of the penises with the females’ paragenital organs

We used µCT scans of the penises and of the paragenital organs of S. ovinae (raw data taken from Peinert et al. (2016)), S. hammella, and S. melittae, and digitally inserted the penises into the paragenital organs with VGStudiomax 2.0.5 (Volume Graphics, Heidelberg, Germany). Doing so illustrated that the size and shape of a specific penis fits only with the paragenital organ of conspecific females (Fig. 7).

Figure 7 Pairs of female cephalothoraces (outer cuticle of the cephalothorax removed) and male penises of Stylops hammella (A), Stylops ovinae (B) and Stylops melittae (C).

Paragenital organs highlighted in ochre, penises in red. S. ovinae was scanned in copula (Peinert et al., 2016). The penises of S. hammella and S. melittae were virtually inserted into the corresponding conspecific female paragenital organs to assess the fit of these genital structures.

Geometric morphometrics

By performing Generalised Procrustes analysis of the morphometric data of S. ovinae penises, we calculated a mean centroid size of 1,829.35, with a standard deviation of 84.88. The coefficient of variation of 4.64% describes the extent of variation of S. ovinae penises with respect to centroid size. Based on a 95% coincidence interval, one outlier was found among the S. ovinae data (Fig. 8B).

The first three axes of the principal components analysis ordination plot explain 64.1% of the sample shape variation of S. ovinae penises (Fig. 8C). PC1 explains 32.1% of the shape variation. From negative to positive PCA values, the dorsal spine shifts towards the base of the penis, resulting in a slightly s-shaped or in a non-s-shaped back of the acumen’s dorsal outline compared to penises related to lower values. Furthermore, the posterior bulge is shifted ventrad towards higher values. PC2 accounts for 20.4% of the sample shape variation. Moving from negative to positive PCA values, shape changes are expressed by a slightly shorter acumen and a sharper ventral angle of the dorsal spine. PC3 describes 11.6% of the total sample shape variation of S. ovinae and relates to a more prominent posterior bulge and slightly elongated posterior spine for positive values.

Figure 8 Geometric morphometric analysis of penises of Stylops ovinae (red) and Xenos vesparum (yellow).

(A) Lateral view of all investigated penises of both species. (B) Shape spaces of penises of both species visualised with the three most representative principal components (see Supplemental for interactive 3D version). (C) and (D) Shape change along the PC axes of S. ovinae (C) and X. vesparum (D). Darker colours represent lower PC values, while brighter colours outline higher PC values.

Generalised Procrustes analysis of morphometric data of X. vesparum penises resulted in a mean centroid size of 1,243.18 with a standard deviation of 45.08. Regarding centroid size, the variation is described by coefficient of variation of 3.6%. As in S. ovinae, an outlier was identified among the data of X. vesparum based on a 95% confidence interval (Fig. 8B).

The first three principal components of the PCA explain 64.9% percent of the overall sample shape variation (Fig. 8D). With a proportion of 38.7% of sample shape variation, negative PC1 values associated with penises having a strongly curved posterior edge straightened towards positive values. The shape variation also influences the dorsal curvature of the acumen, being less curved with positive values than with negative ones. PC2 accounts for 14.1% of total sample shape variation. Compared to negative values, positive values are related to a slightly more downward tilted tip of the acumen for positive PCA values. PC3 describes 12.1% of the sample shape variation. Going from negative to positive, the dorsal spine shifts more dorsad. This appearance leads also to a more strongly bent dorsal curvature of the acumen in X. vesparum.

Discussion

Increased thickness and sclerotization of the insect cuticle are common adaptations to reduce the risk of sexual wounding (Merritt, 1989; Baer & Boomsma, 2006; Kamimura, 2007; Rönn, Katvala & Arnqvist, 2007; Lange et al., 2013; Dougherty & Simmons, 2017; Matsumura et al., 2017). In S. ovinae and X. vesparum, the cuticle of females is weakly sclerotised and uniformly rich in resilin. Most notably, however, it is also three times thicker at the wounding site than at control sites in close spatial proximity. Note that the endoparasitic females are enclosed by the exuviae of the secondary and tertiary larval stage, and that the former is strongly sclerotised, thinning only at the birth opening. The males can only insert their penis at the strongly thinned exuvia of the birth opening. The remaining exuvia of the secondary larva is not suitable for male penetration. Michels, Gorb & Reinhardt (2015) considered bed bugs as an example of a species in which the females have evolved tolerance against traumatic mating. This is due to higher proportions of resilin in the females’ spermalege compared to surrounding areas of the abdomen. The flexibility and resilience of resilin allow efficient sealing of the cuticle after penetration and help reducing loss of haemolymph (Michels, Gorb & Reinhardt, 2015). At the same time, the forces required to pierce the cuticle of the spermalege are notably lower than those at control sites. The high proportion of resilin in the cuticle of the spermalege has been consequently interpreted as a response to traumatic mating, as it facilitates piercing by the male genitalia. The increased thickness of the integument in Strepsiptera requires a higher force to pierce the cuticle at the penetration sites (compared to control sites). This result could indicate that Strepsiptera have evolved resistance rather than tolerance against traumatic insemination. However, Peinert et al. (2016) found S. ovinae males to be consistently able to penetrate their female partners within seconds, indicating that the specific structure of the female cuticle does not seem to hamper penetration by males. We consequently favour an alternative interpretation, namely that the thickened cuticle is also a tolerance trait. As the weakly sclerotised integument of female strepsipterans is already rich in resilin, further increasing the amount of this protein in the integument may not help to mitigate the male-inflicted trauma on the female’s integument. However, increased thickness of the body wall could have such an effect. Potential positive effects of a thickened cuticle could be a reduced risk of integument rupture, improved sealing of the copulation wounds, and reduction of haemolymph loss.

We demonstrated that the sex pheromone emitted by S. ovinae females attracts not only conspecific males, but also males of two additional congeneric species. In contrast, females of X. vesparum attracted only conspecific males. The sympatric occurrence of congeneric species is much more common in Stylops than in Xenos, whose species do not possess a paragenital organ (Kinzelbach, 1971). In the Western Palearctic, only two species of Xenos are described (X. vesparum and Xenos zavattarii (Pierce, 1911) (Benda et al., 2022), the latter only documented from Tripoli, Libya). In contrast, 32 different species of Stylops are currently documented and are accepted as valid species in the Western Palaearctic, and many of them occur in sympatry (Straka, Jůzová & Nakase, 2015). Male Stylops are known to hatch in synchronised masses during a few days in late winter/early spring (Grabert, 1953; Lauterbach, 1954; Tolasch, Kehl & Dötterl, 2012; Lagoutte et al., 2013). This situation most likely leads to copulation of one female with multiple males and to increased interspecific competition. In contrast to males of Stylops, hatching of males of Xenos is not synchronised. They are released over a period from mid-July until mid-August (Hughes et al., 2004). Competition between males should consequently be comparatively low. However, our attraction experiments showed that the elongated flight period of X. vesparum does not mean that males of this species necessarily encounter females in lower frequency than in S. ovinae. In both species, we found multiple males having been attracted by females in short periods of time, even simultaneously. Hughes et al. (2004) and Beani et al. (2018) also observed that several males of X. vesparum were attracted to X. vesparum females at the same time. We consequently hypothesise that in S. ovinae, interspecific competition and intraspecific competition for females are relatively high, whereas in X. vesparum, intraspecific competition is prevalent across most of the distribution range.

Kathirithamby et al. (2015) speculated that females of Strepsiptera produce species-specific pheromones to entice conspecific and to exclude heterospecific males. The sex pheromones are produced by the Nassonow’s glands, which open on the ventral surface of the brood canal in the female’s cephalothorax (Dallai et al., 2004). As we found that the female sex pheromone of S. ovinae attracts heterospecific males, another mechanism is (or additional mechanisms are) likely in place to reduce the chance of heterospecific mating. In his review on the rapid divergent evolution of sexual morphology, Eberhard (2004b) stated that species-specific traits in males are typically present when specific contact organs exist in females. Males among different Stylops species differ strongly in the shape and the size of their intromittent organ (Kinzelbach, 1971; Kinzelbach, 1978) (Figs. 4A–4E). The size of their penises positively correlates with the size of the conspecific female paragenital organ, at least in all three investigated species. In contrast, the penises in Xenos species differ only slightly in size and shape (Kinzelbach, 1971; Kifune & Maeta, 1975; Kifune & Maeta, 1985; Kifune, 1986; Kathirithamby & Hughes, 2006; Nakase & Kato, 2013; Cook, Mayorga-Ch & Sarmiento, 2020) (Figs. 4F–4I). We therefore hypothesise that the paragenital organ in Stylops evolved as a species-specific contact organ for copulation. The proximal part and the acumen (Fig. S3) of the penises vary in width and length among species (Fig. 4). Our mating experiments on S. ovinae showed that only conspecific males are able to mate successfully. Therefore, we conclude that interspecific mating is prevented by structural differences of the paragenital organ of Stylops females. Hence the paragenital organ, in combination with the penis shape, likely functions as a prezygotic barrier between different species. This barrier is apparently missing in the genus Xenos, possibly because heterospecific mating does not occur or is rare for other reasons.

It remains to be investigated whether intraspecific sexual conflict may further accelerate the evolution of the genital morphology within the two sexes of Stylops. Intersexual conflict should increase variation within a population. Therefore, we initially hypothesised that the variation of the penis morphology is larger in S. ovinae than in X. vesparum. However, we found no difference in the magnitude of interspecific variation between the two species, and the males’ intromittent organs of S. ovinae and X. vesparum vary to a similar extent (approximately 4% in centroid size). Therefore, our results do not support this interpretation. However, they do not rule out intraspecific sexual conflict either, as our assumption of intraspecific sexual conflict restricted to species of Stylops could have been wrong.

Conclusions

The results of our study suggest that female strepsipterans of the genera Stylops and Xenos have likely evolved tolerance traits against traumatic insemination. This is achieved by thickening of the uniformly resilin-rich integument at the site of penetration. In contrast, female bed bugs achieved tolerance by incorporating the elastomeric protein resilin at the site of penetration only. This thickening in Stylops and Xenos does not seem to have any negative effect on the male partner, which appears to be always able to pierce the penetration site within seconds. Whether or not mating behaviour and morphological diversity of male penises and female paragenital organs are directly correlated to mating success of either sex is the subject of future research. However, we predict that tolerance traits in the context of traumatic insemination are more widespread in insects than currently assumed.

Supplemental Information

Supplemental Information 1 Female cephalothoraces of Stylops ovinae from mating experiments with males of Stylops melittae described in this study.

(A, E, F) Female Stylops ovinae without mating sign after allospecific mating attempts. (B, C, D) Female Stylops ovinae with injurys from allospecific mating attempts. Abbreviation: ms – mating sign.

Click here for additional data file.

Supplemental Information 2 Exemplary Penetration force curves for each of the four tested scenarios.

(A) Control site of Stylops ovinae. (B) Wounding site of Stylops ovinae (C) Control site of Xenos vesparum. (D) Wounding site of Xenos vesparum. Star indicates penetration of either control or wounding site.

Click here for additional data file.

Supplemental Information 3 Scanning electron micrographs of the penises of Stylops ovinae, Xenos vesparum and the tip of the microneedle used in the penetration experiments.

(A) Penis of Stylops ovinae, lateral view. (B) Penis of Xenos vesparum, lateral view. (C) Tip of the microneedle. Abbreviations: ac – acumen, ds – dorsal spine, fs – frontal spine.

Click here for additional data file.

Supplemental Information 4 Phylogenetic relationships of COI barcode nucleotide sequences from species of the genus Stylops and of closely related genera.

COI nucleotide sequences published by Jůzová, Nakase & Straka (2015) served as references. Sample IDs starting with Jen and followed by a number indicate male Stylops samples studied in the present investigation. The phylogeny was inferred with the neighbour-joining method and applying the Tamura-Nei substitution model. Values at branches indicate bootstrap support over 50%.

Click here for additional data file.

Supplemental Information 5 Movie clips illustrating the conducted mating experiments.

Two exemplary movie clips of male Stylops of two different species (Stylops hammella, S. melittae) unsucssesfully copulate with allospecific females of S. ovinae.

Click here for additional data file.

Supplemental Information 6 Raw measurements of the micro-indentation experiments.

Click here for additional data file.

Supplemental Information 7 Interactive shape spaces of penises of Stylops ovinae (red) and Xenos vesparum (yellow) visualised with the three most representative principal components.

Click here for additional data file.

We thank Eberhardt Holtappels (Langerwehe) for his help with collecting stylopised bees and Irmgard and Hans Muth (Mettenheim) for the permission to collect wasps on their private property. We also thank Daniel Tröger (Jena) for collecting infested bees and wasps. Furthermore, we thank Jakub Straka (Praha) for providing material for this study, Adrian Richter (Jena) for performing µCT scans, Alexander Stoessel for providing access to the equipment at the Max-Planck-Institut für Menschheitsgeschichte (Jena), and Marcus Zuber and Tomás Farago (both KIT) for their help at the beamline and with the reconstruction of synchrotron µCT data. We acknowledge the KIT light source for provision of instruments at their beamlines. We also thank the Institute for Beam Physics and Technology (IBPT) for the operation of the storage ring, the Karlsruhe Research Accelerator (KARA). Furthermore, our thanks go to Brendon Boudinot for helpful input to the manuscript. Finally, we thank Joseph Gillespie, Laura Beani, and Leon Lounibos for their helpful suggestions, which helped improving the manuscript.

Additional Information and Declarations

Competing Interests

Author Contributions

Data Availability

Stanislav Gorb is an Academic Editor for PeerJ.

Kenny Jandausch conceived and designed the experiments, performed the experiments, analyzed the data, prepared figures and/or tables, authored or reviewed drafts of the article, and approved the final draft.

Jan Michels conceived and designed the experiments, performed the experiments, analyzed the data, prepared figures and/or tables, authored or reviewed drafts of the article, and approved the final draft.

Alexander Kovalev conceived and designed the experiments, performed the experiments, analyzed the data, authored or reviewed drafts of the article, and approved the final draft.

Stanislav N. Gorb conceived and designed the experiments, performed the experiments, analyzed the data, authored or reviewed drafts of the article, and approved the final draft.

Thomas van de Kamp conceived and designed the experiments, performed the experiments, analyzed the data, authored or reviewed drafts of the article, and approved the final draft.

Rolf Georg Beutel conceived and designed the experiments, analyzed the data, authored or reviewed drafts of the article, and approved the final draft.

Oliver Niehuis conceived and designed the experiments, performed the experiments, analyzed the data, authored or reviewed drafts of the article, and approved the final draft.

Hans Pohl conceived and designed the experiments, performed the experiments, analyzed the data, prepared figures and/or tables, authored or reviewed drafts of the article, and approved the final draft.

The following information was supplied regarding data availability:

The raw measurements of the micro-indentation experiments are available in the Supplemental Files.

MicroCT Data are available at Morphosource:

https://www.morphosource.org/projects/000430676

- Copulating pair of Stylops ovinae CT Image Series; Media 000431013; MorphoSource DOI 10.17602/M2/M431013.

- Female cephalothorax of Stylops hammmella CT Image Series; Media 000430994; MorphoSource DOI 10.17602/M2/M430994.

- Female cephalothorax of Stylops melittae CT Image Series; Media 000431002; MorphoSource DOI 10.17602/M2/M431002.

- Penis of Stylops hammmella CT Image Series; Media 000430950; MorphoSource DOI 10.17602/M2/M430950.

- Penis of Stylops melittae CT Image Series; Media 000430981; MorphoSource DOI 10.17602/M2/M430981

18 Penises of Stylops ovinae surface models:.

- pmj:Strep_42/01: Media 000432425; MorphoSource DOI 10.17602/M2/M432425.

- pmj:Strep_42/02: Media 000432431; MorphoSource DOI 10.17602/M2/M432431.

- pmj:Strep_42/03: Media 000432437; MorphoSource DOI 10.17602/M2/M432437.

- pmj:Strep_42/04: Media 000432442; MorphoSource DOI 10.17602/M2/M432442.

- pmj:Strep_42/05: Media 000432448; MorphoSource DOI 10.17602/M2/M432448.

- pmj:Strep_42/06: Media 000432454; MorphoSource DOI 10.17602/M2/M432454.

- pmj:Strep_42/08: Media 000432460; MorphoSource DOI 10.17602/M2/M432460.

- pmj:Strep_42/09: Media 000432466; MorphoSource DOI 10.17602/M2/M432466.

- pmj:Strep_42/10: Media 000432472; MorphoSource DOI 10.17602/M2/M432472.

- pmj:Strep_42/11: Media 000432478; MorphoSource DOI 10.17602/M2/M432478.

- pmj:Strep_42/12: Media 000432484; MorphoSource DOI 10.17602/M2/M432484.

- pmj:Strep_42/13: Media 000432489; MorphoSource DOI 10.17602/M2/M432489.

- pmj:Strep_42/14: Media 000432495; MorphoSource DOI 10.17602/M2/M432495.

- pmj:Strep_42/15: Media 000432501; MorphoSource DOI 10.17602/M2/M432501.

- pmj:Strep_42/16: Media 000432507; MorphoSource DOI 10.17602/M2/M432507.

- pmj:Strep_42/17: Media 000432512; MorphoSource DOI 10.17602/M2/M432512.

- pmj:Strep_42/19: Media 000432518; MorphoSource DOI 10.17602/M2/M432518.

- pmj:Strep_42/22: Media 000432529; MorphoSource DOI 10.17602/M2/M432529.

17 Penises of Xenos vesparum surface models:

- pmj:Strep_43/02: Media 000432535; MorphoSource DOI 10.17602/M2/M432535.

-pmj:Strep_43/03: Media 000432541; MorphoSource DOI 10.17602/M2/M432541.

-pmj:Strep_43/04: Media 000432547; MorphoSource DOI 10.17602/M2/M432547.

-pmj:Strep_43/05: Media 000432553; MorphoSource DOI 10.17602/M2/M432553.

-pmj:Strep_43/06: Media 000432559; MorphoSource DOI 10.17602/M2/M432559.

-pmj:Strep_43/07: Media 000432565; MorphoSource DOI 10.17602/M2/M432565.

-pmj:Strep_43/08: Media 000432571; MorphoSource DOI 10.17602/M2/M432571.

-pmj:Strep_43/09: Media 000432577; MorphoSource DOI 10.17602/M2/M432577.

-pmj:Strep_43/10: Media 000432583; MorphoSource DOI 10.17602/M2/M432583.

-pmj:Strep_43/11: Media 000432589; MorphoSource DOI 10.17602/M2/M432589.

-pmj:Strep_43/12: Media 000432595; MorphoSource DOI 10.17602/M2/M432595.

-pmj:Strep_43/13: Media 000432601; MorphoSource DOI 10.17602/M2/M432601.

-pmj:Strep_43/14: Media 000432607; MorphoSource DOI 10.17602/M2/M432607.

-pmj:Strep_43/15: Media 000432613; MorphoSource DOI 10.17602/M2/M432613.

-pmj:Strep_43/16: Media 000432619; MorphoSource DOI 10.17602/M2/M432619.

-pmj:Strep_43/17: Media 000432625; MorphoSource DOI 10.17602/M2/M432625.

-pmj:Strep_43/18: Media 000432631; MorphoSource DOI 10.17602/M2/M432631.

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
