# Peer review of "Have female twisted-wing parasites (Insecta: Strepsiptera) evolved tolerance traits as response to traumatic penetration?"

_PeerJ, doi:10.7717/peerj.13655_

## Round 0.1 · original submission · Major Revisions

Dear Dr. Jandausch and colleagues:

Thanks for submitting your manuscript to PeerJ. I have now received three independent reviews of your work, and as you will see, the reviewers raised some minor concerns about the manuscript. Despite this, these reviewers are highly optimistic about your work and the potential impact it will lend to research on strepsipteran biology. Thus, I encourage you to revise your manuscript, accordingly, taking into account all of the concerns raised by the reviewers.

While the concerns of the reviewers are relatively minor, this is a major revision to ensure that the original reviewers have a chance to evaluate your responses to their concerns.

Please note that reviewers 1 and 2 have included marked-up versions of your manuscript.

Importantly, please add the references considered by the reviewers to be important for placing your study within the full context of the system. Please also ensure that access to all supporting material is available.

I look forward to seeing your revision, and thanks again for submitting your work to PeerJ.

Good luck with your revision,

Best,

-joe

·

Basic reporting

BASIC REPORTING Clear, professional English language used throughout. Relevant literature is lacking. The structure conforms to PeerJ standards. High quality figures, well labelled. Raw data supplied.

Experimental design

EXPERIMENTAL DESIGN Original primary research within Scope of the journal. Research question in Stylops well defined, not equally in Xenos. High technical & ethical standard. Methods detailed.

Validity of the findings

VALIDITY OF THE FINDINGS High impact and novelty. Literature is clearly stated for Stylops but incomplete for Xenos. Data are robust and statistically controlled. Conclusions support results.
General comments, criticisms and suggestions
INTRODUCTION
Abstract is perfect, while Introduction could be shortened to be more effective. For example, omit any information on Mengenillidae (lines 50-52, 77-78, 82-83, 111-113).
Given the topic – traumatic insemination - I suggest avoiding the term “copulate” (76), “copulation” (90), “copulate” (311), sensu Eberhard (1995, p.69): “copulation (genitalic coupling of male and female) is not equivalent to insemination.”
89-90: “the secondary copulation opening”: unclear. Do you mean that this opening “is used in most species for insemination” (93-94)? My suggestion is to quote and discuss a study on Xenos vesparum in which this alternative sperm routes have been described, besides traumatic insemination. This study describes - with scanning and transmission electron microscopy - two routes by which sperm cells can reach the hemocoel, hypodermic insemination and/or extragenital insemination via ventral canal (compare Fig.5 with Fig.2B in this ms).
Beani, L., Giusti, F., Mercati, D., Lupetti, P., Paccagnini, E., Turillazzi, S., & Dallai, R. (2005). Mating of Xenos vesparum (Rossi) (Strepsiptera, Insecta) revisited. Journal of Morphology, 265(3), 291-303.



MATERIAL AND METHODS
161-163: preference of Xenos vesparum for Campsis radicans has been described in detail in a recent study. Add this reference, please, and briefly comment the altered feeding of parasitized wasps. I wonder if Andrena vaga bees change their behavior if parasitized as well as X. vesparum, for example if they aggregate and thus facilitate parasite’s mating.
Beani, L., Cappa, F., Manfredini, F., & Zaccaroni, M. (2018). Preference of Polistes dominula wasps for trumpet creepers when infected by Xenos vesparum: A novel example of co-evolved traits between host and parasite. PloS one, 13(10), e0205201.
I appreciated the integrated methods used in this study, both observational and experimental: confocal laser scanning microscope, micro-indentation experiments, scanning electron microscopy, attraction experiments, specimen identification, mating experiments, X-ray computed tomography, 3D-reconstruction. I wonder if reading this beautiful research could be facilitated by transferring the very technical paragraphs “geometric morphometrics, landmarks and image processing” in Supplementary Methods, since “the males’ intromittent organs of S. ovinae and X. vesparum vary to a similar extent.” (549-552).
RESULTS and DISCUSSION
The main results of this study are the thicker resilin-rich integument and the higher forces required to penetrate in areas where traumatic insemination takes place rather than in control areas, either in Stylops or in Xenos, although the latter is less sclerotized. Co-evolution of the penises with the females’ paragenital organs in Stylops is a further intriguing result, as well as the alternative hypothesis that Strepsiptera have evolved tolerance rather than resistance against traumatic insemination: lower risk of integument rupture, sealing of the copulation wounds and reduced haemolymph loss (504-505).

As regards Attraction and Mating experiments in Stylops, you could quote similar field observations carried out in X. vesparum: five volant X. vesparum males, one mating on a leaf 20 cm from an aggregation and seven males close to a caged receptive female.
Hughes, D. P., Kathirithamby, J., Turillazzi, S., & Beani, L. (2004). Social wasps desert the colony and aggregate outside if parasitized: parasite manipulation? Behavioral Ecology, p.1042.
Again, in lab we described the direct approach by a male towards a receptive female, host interference by wings and legs movements, quick mating, and subsequent death of the male.
Beani et al. (2005). Mating of Xenos vesparum (Rossi) (Strepsiptera, Insecta) revisited (p.297)
Moreover, wasps parasitized by one or two X. vesparum females, individually enclosed into vials covered with a mesh, attracted male parasites inside the vials.
Beani et al. (2018). Preference of Polistes dominula wasps for trumpet creepers when infected by Xenos vesparum.

Another original information is the sympatric occurrence of congeneric species, which increased interspecific competition (506-517) in Stylops but not in X. vesparum, released “over a period of several weeks” (518-519). More exactly, males emerge from mid-July until mid-August, the peak of the mating period, as indicated by the maximum number of empty puparia.
Hughes et al. (2004). Social wasps desert the colony and aggregate outside if parasitized: parasite manipulation? Behavioral Ecology. Fig.2, p.1039.
According with Kathyrithamby (2015), females of Strepsiptera produce species-specific pheromones to attract conspecific and to exclude heterospecific males (526-527). Although further prezygotic barriers may reduce heterospecific mating (527-545), I think that you could describe the Nassonow’s gland, which opens behind the brood canal up to the neck region. Dallai, R., Lupetti, P., Giusti, F., Mercati, D., Paccagnini, E., Turillazzi, S., Beani, L., Kathirithamby, J. (2004). Fine structure of the Nassonow’s gland in the neotenic endoparasitic of female Xenos vesparum (Rossi) (Strepsiptera, Insecta). Tissue and Cell, p.214.

Comments by Laura Beani

·

Basic reporting

good; see attached file for suggested English improvements

Experimental design

good, answering more questions than the title suggests.....see attached review

Validity of the findings

Discussion is cautious....at the discretion of the authors could be expanded.

Additional comments

none

Reviewer 3 ·

Basic reporting

NA

Experimental design

NA

Validity of the findings

NA

Additional comments

This is a rare case that I had no specific comments. The paper was interesting and similar results as in other systems.

---

## Round 0.2 · accepted · Accept

Dear Dr. Jandausch and colleagues:

Thanks for revising your manuscript based on the concerns raised by the reviewers. I now believe that your manuscript is suitable for publication. Congratulations! I look forward to seeing this work in print, and I anticipate it being an important resource for groups studying strepsipteran biology. Thanks again for choosing PeerJ to publish such important work.

Best,

-joe

---

## Author Rebuttal · Round 0.2

**FRIEDRICH-SCHILLER-UNIVERSITÄT JENA**

**Institut für Zoologie und Evolutionsforschung**

mit Phyletischem Museum, Ernst-Haeckel-Haus und Biologiedidaktik

PD Dr. Hans Pohl

Erbertstrasse 1
07743 Jena

| Telefon: | 0 36 41 9-49156 |
| Telefax: | 0 36 41 9-49142 |
| E-Mail: | hans.pohl@uni-jena.de |
| | www.entomology.uni-jena.de |

Universität Jena · Institut für Zoologie und Evolutionsforschung

To

Dr. Joseph Gillespie

May 23, 2022

Dear Dr. Gillespie:

Thank you very much for your efforts with handling our manuscript and for your comments and suggestions to improve our study. We individually address the comments of the reviewers below. In some cases, where we did not agree with the reviewers suggestions, we provide arguments for our point of view.

**Detailed responses to the comments** (reviewers' comments in red, authors' response in black):

**Reviewer: Laura Beani**

"Abstract is perfect, while Introduction could be shortened to be more effective. For example, omit any information on Mengenillidae (lines 50-52, 77-78, 82-83, 111-113). "

We do not agree with this comment. We believe that the information provided on Mengenillidae gives a more complete picture of the Strepsiptera. Furthermore, we would refrain from shortening the introduction, as provided information is important for understanding our manuscript in its details.

"Given the topic – traumatic insemination - I suggest avoiding the term "copulate" (76), "copulation" (90), "copulate" (311), sensu Eberhard (1995, p.69): "copulation (genitalic coupling of male and female) is not equivalent to insemination.""

Many thanks for the suggestion. We are aware that copulation and insemination are not equivalent. We changed the manuscript text accordingly.

"89-90: "the secondary copulation opening": unclear. Do you mean that this opening "is used in most species for insemination" (93-94)? My suggestion is to quote and discuss a study on Xenos vesparum in which this alternative sperm routes have been described, besides traumatic insemination. This study describes - with scanning and transmission electron microscopy - two routes by which sperm cells can reach the hemocoel, hypodermic insemination and/or extragenital insemination via ventral canal (compare Fig.5 with Fig.2B in this ms).

[Figure]

Beani, L., Giusti, F., Mercati, D., Lupetti, P., Paccagnini, E., Turillazzi, S., & Dallai, R. (2005). Mating of Xenos vesparum (Rossi) (Strepsiptera, Insecta) revisited. Journal of Morphology, 265(3), 291-303. "

Thank you for pointing this out. We did not want to go into detail on the different insemination openings of the Corioxenidae. However, as this has led to confusion, we included it (l 99–100). In lines 110–113, we have added the requested comment and the corresponding citation from Beani et al. (2005) on the alternative sperm route.

"161-163: preference of *Xenos vesparum* for *Campsis radicans* has been described in detail in a recent study. Add this reference, please, and briefly comment the altered feeding of parasitized wasps. I wonder if *Andrena vaga* bees change their behavior if parasitized as well as *X. vesparum*, for example if they aggregate and thus facilitate parasite's mating.
Beani, L., Cappa, F., Manfredini, F., & Zaccaroni, M. (2018). Preference of *Polistes dominula* wasps for trumpet creepers when infected *by Xenos vesparum*: A novel example of co-evolved traits between host and parasite. PloS one, 13(10), e0205201."

Thank you very much for this advice. We forgot to mention this behaviour. We added the citation and a short comment at this point as suggested. In contrast to parasitised *Polistes dominula*, aggregation is not known from parasitised *Andrena vaga*. However parasitised *A. v*aga hatch before the uninfected individuals.

"I appreciated the integrated methods used in this study, both observational and experimental: confocal laser scanning microscope, micro-indentation experiments, scanning electron microscopy, attraction experiments, specimen identification, mating experiments, X-ray computed tomography, 3D-reconstruction. I wonder if reading this beautiful research could be facilitated by transferring the very technical paragraphs "geometric morphometrics, landmarks and image processing" in Supplementary Methods, since "the males' intromittent organs of *S. ovinae* and *X. vesparum* vary to a similar extent." (549-552)."

Thank you for the compliment. We initially hypothesised that the variation of the penis morphology is larger in *S. ovinae* than in *X. vesparum*. This has not been confirmed and could only be verified by geometric morphometrics. Therefore, we do not follow the suggestion to transfer this section to the Supplement. The section Image processing takes only three lines, and we therefore leave it in the chapter Material and Methods as well.

"As regards Attraction and Mating experiments in *Stylops*, you could quote similar field observations carried out in *X. vesparum*: five volant *X. vesparum* males, one mating on a leaf 20 cm from an aggregation and seven males close to a caged receptive female.
Hughes, D. P., Kathirithamby, J., Turillazzi, S., & Beani, L. (2004). Social wasps desert the colony and aggregate outside if parasitized: parasite manipulation? Behavioral Ecology, p.1042.
Again, in lab we described the direct approach by a male towards a receptive female, host interference by wings and legs movements, quick mating, and subsequent death of the male.

[Figure]

Beani et al. (2005). Mating of *Xenos vesparum* (Rossi) (Strepsiptera, Insecta) revisited (p.297) Moreover, wasps parasitized by one or two *X. vesparum* females, individually enclosed into vials covered with a mesh, attracted male parasites inside the vials.

Beani et al. (2018). Preference of *Polistes dominula* wasps for trumpet creepers when infected by *Xenos vesparum*."

Many thanks for these suggestions. We have added the two references (Hughes et al. 2004 and Beani et al. 2018) to the discussion (lines 630–631). We have not cited the work of Beani et al. (2005) at this point, as only one male was attracted by the female.

"Another original information is the sympatric occurrence of congeneric species, which increased interspecific competition (506-517) in Stylops but not in X. vesparum, released "over a period of several weeks" (518-519). More exactly, males emerge from mid-July until mid-August, the peak of the mating period, as indicated by the maximum number of empty puparia.

Hughes et al. (2004). Social wasps desert the colony and aggregate outside if parasitized: parasite manipulation? Behavioral Ecology. Fig.2, p.1039."

Thank you, we have added this.

"According with Kathyrithamby (2015), females of Strepsiptera produce species-specific pheromones to attract conspecific and to exclude heterospecific males (526-527). Although further prezygotic barriers may reduce heterospecific mating (527-545), I think that you could describe the Nassonow's gland, which opens behind the brood canal up to the neck region. Dallai, R., Lupetti, P., Giusti, F., Mercati, D., Paccagnini, E., Turillazzi, S., Beani, L., Kathirithamby, J. (2004). Fine structure of the Nassonow's gland in the neotenic endoparasitic of female Xenos vesparum (Rossi) (Strepsiptera, Insecta). Tissue and Cell, p.214."

Thank you very much for this advice. We have added a sentence on the Nassonow's glands and also labelled the Nassonow's glands in Figure 2.

**Reviewer: Leon Lounibos**

"This MS represents a creative, multifaceted contribution on the reproductive biology of a poorly known but fascinating order of insects. An attractive feature of the submission is its melding of high-tech instrumental resolution, appropriate analytic methods, and standard entomological procedures to tease out conclusions about both female and male sexual behaviors and evolution of the Strepsiptera."

Thank you for the compliment.

"The title of the MS understates its contents, i.e., the demonstration of an extra layer of cuticle where the male intromittent organ penetrates the female cephalothorax represents only one portion of the Results. The Discussion is relatively conservative, e.g., avoiding controversial topics that arise

in the Introduction and Results, such as lock and key (mechanism) and coevolution. The observation of this conservative approach is not a criticism, but the MS might attract a broader readership through discussion of the (potentially) controversial terms of sexual selection."

We discussed an earlier draft of our manuscript with some colleagues in advance and received feedback that we should be rather cautious with our statements. We followed that advice. In this respect, we would like to keep the title and also stay with our relatively conservative discussion.

"l.94-95: The meaning of "The female represents a functional unit of the exuviae……"is unclear. I suggest that you rephrase to indicate that the exuviae of second and third larval instars are incorporated into the female exoskeleton."

We rephrased this sentence and hope that the meaning is now clearer.

"Fig.1: A measurement scale in needed for each panel (A-D)."

We added a scale bar to each panel.

"Fig.2: To the legend, add 'female' cephalothorax."

Thank you for this comment. We added "female".

"l.114: "intraspecific interbreeding" seems (to me) to be a tautology; please rephrase"

That is correct, thank you. We deleted "Intraspecific".

"l.160: "to prevent males from hatching"….indicate males of which species, e.g., A. vega or S.ovinae."

We added the species (*i.e.*, *S. ovinae*).

"l.172: an 'aerarium' was a public treasury in ancient Rome; perhaps use "cage", instead?"

Thank you for pointing out that an aerarium was a public treasury in ancient Rome. However, "aerarium" is the official name of the manufacturer for this type of insect cages. We have therefore not changed the term, but have consistently put it in inverted commas.

"l.212: were females alive during the 'micro-indentation experiments'?"

Yes, the females were alive to ensure that the cuticle properties were not changed by the addition of killing agents such as ethyl acetate. We added "living" at this point.

"l.217: the µm notation is repeated twice in succession."

We deleted this duplication.

"l.218: Include variance terms and 'n' with the 4.0 µm and 0.7 µm measurements."

We initially measured only one penis tip at a time. Now we have re-measured three tips each and added the variance values (n, min., max.).

"l.233-247: consider broadening the Statistical analysis section to include the PCA. (If necessary, this section could be transposed to later in the Materials and Methods.)"

That is an excellent suggestion. We included PCA in the section "Statistical analysis", and transferred the whole section to a later place in the chapter "Material and Methods".

"l.243: I suspect that the 'pairwise Wilcox test' should be the Wilcoxon pairwise comparison."

We replaced the term as suggested.

"l.256-258 & l.262-264: If the methods for attracting male Stylopodia in the field and for cooling live males to keep them vigorous for experiments have been published previously, please provide citations."

To our knowledge, this method has not yet been used for *Stylops*. However, Beani et al. (2018) have attracted males of *Xenos vesparum* in the field. We have added a sentence to this.

"l.311: change 'try' to "tried'"

Thank you, we changed the text.

"l.357: 'decimated' means annihilated or obliterated; if you meant reduced to a decimal, try "decimalized""

We replaced "decimated" with "the polygons were reduced".

"l.382: change 'San Jose, USA' to San Jose CA, USA because other states in the USA have cities named San Jose."

Thank you for the kind advice. We changed the text.

"Table 1: in a footnote, explain 'mN' and 'GPa'; numbers of observations should be recorded for each line."

These abbreviations are now explained in the legend and we added the number of observations as suggested.

"Table 2 & l.398-406, l.404-405, l.411-412, l.415-416: It is not clear why both K-W and Wilcoxon paired comparisons results are given for some comparisons. The Kruskal-Wallis (note spelling) test is appropriate only if numbers of groups are three or more."

Thank you for pointing this out. We corrected this at the mentioned positions. p-values in the manuscript and the table are only given from the Wilcoxon paired comparison, as they indicate the statistical significance.

"Fig.6: Legend should explain box plot details, such as the red and yellow dots above or below three plots."

[Figure]

We added the following sentence to the figure legend to explain the box plot details: "Boxes represent the interquartile range between first and third quartiles and the line inside represents the median. Whiskers denote the lowest and highest values within 1.5× interquartile range from the first and third quartiles, respectively. Circles represent outliers beyond the whiskers."

"l.428-429: Is this section meant to apply to both S. ovinae and X. vesparum? (perhaps use species names in the sentence.)"
We added the species names to avoid misunderstanding.

"Fig. 7: Legend should explain 'cephalothoraces of female S. ovinae.'"
We adjusted the legend slightly to point out that each panel shows a pair of a female cephalothorax and a male penis of the same species.

"l.442-447: is coevolution appropriately invoked here? (See, e.g., Tong & Huang 2019)."
Thank you for pointing out the publication by Tong & Huang (2019). The authors propose the following principle for the correct use of the term coevolution: species interaction, reciprocal selection as well as co-phylogenesis. You are right. These three principles do not apply to our use of the term. We have exchanged the term with "co-adaptation".

"Do the authors see this as Sexually Antagonistic Coevolution (SAC) as described by Tataric et al. (2014)?"
Tatarnic et al. (2014: 253) describe SAC as follows: "Under the predictions of sexual conflict, antagonistic traits are expected to evolve in both sexes as males and females struggle for control over reproduction (4, 66). This can lead to sexually antagonistic coevolution (SAC), in which coercive or cost-inducing traits in one sex are met by evolutionary counteradaptations in the other (2, 4, 66, 79) in an ongoing coevolutionary chase, similar to parasite-host coevolution."
We do not think that the co-adaptation of the penises and female paragenitalia of *Stylops* represent an example of sexually antagonistic coevolution. We assume that the paragenital organ in the genus *Stylops* represents a prezygotic mating barrier that prevents heterospecific mating. However, detecting SAC in Strepsiptera (e.g., a shortened life span or fecundity) is extremely difficult or even impossible due to the very short life span of males of only a few hours and the permanently endoparasitic lifestyle of females.

"l.451,465: the appropriate English is coefficient of "variation" not, 'variance'"
Thank you for pointing this out. We changed the text accordingly.

"l.469: change 'straightening' to "straightened""
We changed the text accordingly.

[Figure]

"l.488: remove 'then'"

We removed "then".

"l.498: what is meant by a 'noteworthy' resistance barrier?"

We clarified this and changed it as follows: "…indicating that the specific structure of the female cuticle does not seem to hamper penetration by males."

"l.511: omit "one"2

We removed 'one'.

2l.512: In English, the capitol of Libya is "Tripoli""

We changed it to 'Tripoli'.

"l.515: insert "a" after 'during' and before 'few'"

We corrected the sentence.

"l. 517: is polygamy documented in Strepsiptera?"

Polygamy has not yet been documented in Strepsiptera in the wild. However, there are field observations showing that several males of *Stylops ovinae* compete for one female (see Figure 8 in Peinert et al. 2016). Peinert et al. (2016) observed multiple copulations of a single female with at least two different males in *S. ovinae* under laboratory conditions. These observations make it very likely that polygamy also occurs in the field, at least in *Stylops*.

"l.564-565: Might the evolution of 'tolerance traits' be an example of coevolution?"

No, we do not think so. The effects of tolerance traits on the fitness of counterparts are often neutral, so they do not trigger a coevolutionary arms race and are therefore not an example of coevolution.

"l. 594-607: 7/8 of the data sets were not available for review, because these will not be released until 'after acceptance'."

We checked this issue. Only the DOI will be provided after acceptance. We made all mentioned data available on Morphosource for the reviewers via the PeerJ Reviewer account. All data can be found in the following project.
https://www.morphosource.org/projects/000430676.

**Reviewer 3:**
No comment was made to improve the manuscript.

[Figure]

We hope that we addressed all comments appropriately. If there are any questions or comments, please do not hesitate to contact us.

Sincerely yours,

(Hans Pohl)